# DFVEdit: Conditional Delta Flow Vector for Zero-shot Video Editing

## Abstract

The advent of Video Diffusion Transformers (Video DiTs) marks a milestone in video generation. However, directly applying existing video editing methods to Video DiTs often incurs substantial computational overhead, due to resource-intensive attention modification or finetuning. To alleviate this problem, we present DFVEdit, an efficient zero-shot video editing method tailored for Video DiTs. DFVEdit eliminates the need for both attention modification and fine-tuning by directly operating on clean latents via flow transformation. To be more specific, we observe that editing and sampling can be unified under the continuous flow perspective. Building upon this foundation, we propose the Conditional Delta Flow Vector (CDFV) – a theoretically unbiased estimation of DFV – and integrate Implicit Cross Attention (ICA) guidance as well as Embedding Reinforcement (ER) to further enhance editing quality. DFVEdit excels in practical efficiency, offering at least 20x inference speed-up and 85% memory reduction on Video DiTs compared to attention-engineering-based editing methods. Extensive quantitative and qualitative experiments demonstrate that DFVEdit can be seamlessly applied to popular Video DiTs (*e.g.*, CogVideoX and Wan2.1), attaining state-of-the-art performance on structural fidelity, spatial-temporal consistency, and editing quality.

## 1 Introduction

In the wave of digitization, video creation has become a dominant form of entertainment. In response, research on controllable video generation holds considerable practical importance. While Video Diffusion Transformer (DiT) models [1–4] have revolutionized video synthesis quality, and DiT-based image editing methods [5–10] have achieved remarkable success, video editing remains challenging in preserving spatiotemporal fidelity. Critically, existing video editing methods do not fully exploit the capabilities of Video DiTs, limiting the potential for high-quality controllable video generation.

Existing video editing techniques mainly follow two paradigms: training-based methods [11–14] and zero-shot methods [15–20]. Given that the former requires resource-intensive finetuning, our work focuses on training-free video editing. For training-free video editing, a high-quality pre-trained base model is cru-

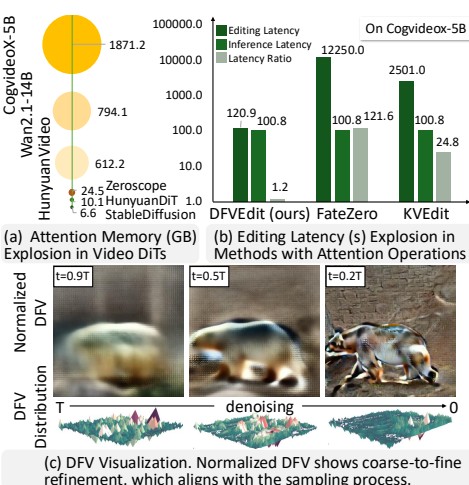

Figure 1: Key insight and motivation.

(a) Attention Memory (GB) Explosion in Video DiTs

(b) Editing Latency (s) Explosion in Methods with Attention Operations

(c) DFV Visualization. Normalized DFV shows coarse-to-fine refinement, which aligns with the sampling process.

cial. Early video editing methods primarily utilized image diffusion models [21, 22], which suffered from temporal inconsistencies due to the lack of capable video diffusion models. These early methods [23, 15, 17, 24] not only had to ensure structural integrity and editing accuracy but also required significant effort to enhance temporal coherence. In contrast, methods [16, 25] based on video diffusion models naturally excel in temporal consistency, leading us to leverage the latest Video DiTs [4, 1, 2] for video editing. Regardless of the type of base models, achieving high fidelity and temporal consistency hinges on attention engineering in most existing methods, including various attention caching and modification techniques. The key to effective attention engineering is that attentions (including keys, queries, and values) contain the spatial-temporal information of the source video, allowing for smooth editing of target regions while preserving the original content's integrity. However, attention mechanisms now consume hundreds of gigabytes of memory (Fig. 1(a)) in Video DiTs [1, 4, 2], a significant increase from previous usage in Unet-based diffusion models [26, 21, 22] and image DiT models [27, 28] at the gigabyte scale. This suggests that traditional attention engineering techniques are incompatible with Video DiTs, creating an urgent need for methods that preserve editing quality while improving computational efficiency.

Motivated by this inefficiency, we shift the focus from attention to input latents and introduce a continuous flow transformation framework for direct video latent refinement. We observe that the standard sampling process in video diffusion models—whether based on Score Matching [29] or Flow Matching [30]—can be unified under a continuous flow perspective. Based on this insight, we demonstrate that editing from the source to the target video naturally forms a time-dependent flow vector field (Fig. 1(c)), which we term the Delta Flow Vector (DFV).

Building upon this foundation, we introduce the Conditional Delta Flow Vector (CDFV) to estimate the flow from source to target latent, incorporating Implicit Cross Attention Guidance (ICA) and Embedding Reinforcement (ER) to further improve editing accuracy. The CDFV in Video DiTs inherently enforces spatial-temporal dependencies while its divergence directly determines update weights. This physically grounded formulation provides two fundamental advantages over approximation-based latent-refinement approaches like DDS [31] and SDS [32]: (1) *theoretical unification* by modeling both sampling and editing from the continuous flow perspective and (2) *computational efficiency* through divergence-determined and hyperparameter-free weights that eliminate heuristic scheduling and overcome low convergence issues inherent to shallow approximations. Moreover, for the seamless application to video editing, we enhanced spatiotemporal coherence by intrinsically avoiding randomness bias while incorporating ICA guidance and ER mechanisms (Fig. 5). Experiments show DFVEdit achieves at least 20× speed-up and 85% memory reduction over attention-engineering-based methods on Video DiTs (*e.g.*, CogVideoX, Wan2.1), while maintaining SOTA performance in fidelity, temporal consistency, and editing quality. Consequently, our approach offers an efficient and versatile solution for zero-shot video editing on Video DiTs.

## 2 Related work

**Video Diffusion Transformer.** Video Diffusion Transformers have evolved from early 3D-UNet-based designs [33, 26, 34, 35] to modern 3D-Transformer-based designs [3]. Advanced models such as Open-Sora [36, 37], CogVideoX [1], HunyuanVideo [2] and Wan [4] have all or part of the following key innovations: replacement of 3D-UNets with scalable 3D-Transformer blocks; integration of cross-attention and self-attention into a unified 3D-full-attention [1, 2]; and adoption of 3D-VAE [1] for spatiotemporal latent compression. Some Video DiTs [27, 4] are combined with Flow Matching [30] while others [1] adopt SDE [29] samplers like DPM-solver [38].

**Image editing on Diffusion Transformer.** With the rise of Diffusion Transformer [3], DiT-based image editing methods [28, 27] have emerged. However, directly applying image editing methods to videos often fails to address temporal consistency and motion fidelity. Additionally, adapting them to Video DiTs introduces extra challenges. Firstly, generalization limitations occur when applying methods [8, 6, 9, 10, 39, 40] that rely on rectified flow [41] or distilled few-step models [42] to Video DiTs that are not combined with rectified flow or distillation techniques. Secondly, efficiency limitations are present for image editing methods [43] that require finetuning. Furthermore, even generalized and efficient methods like DiT4Edit [5] and KVEdit [7], which use attention or key-value caching and modification, still face prohibitive computational costs due to the more massive attention overhead in Video DiTs compared to image DiTs.

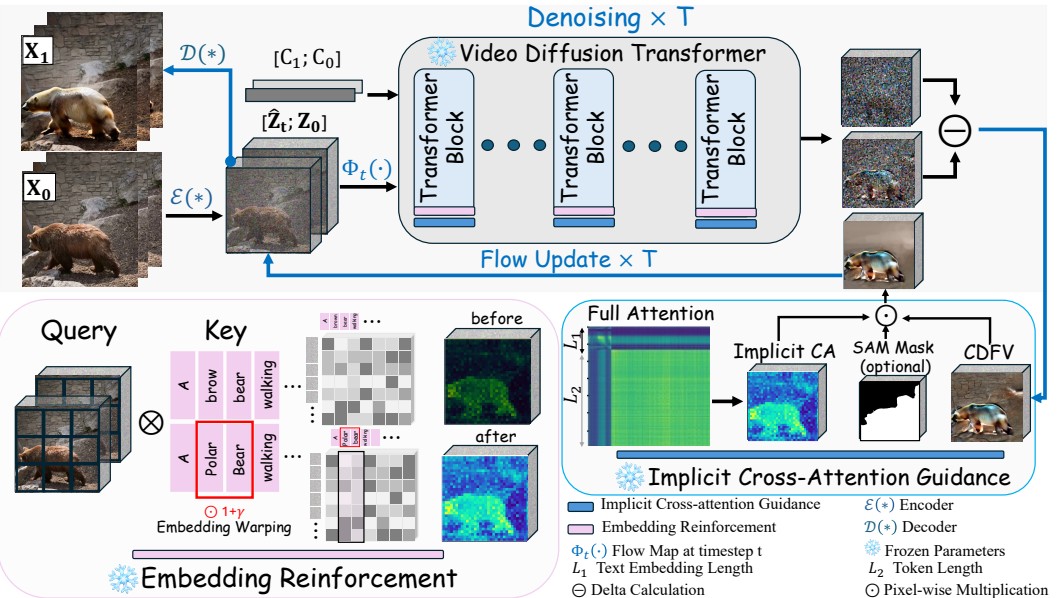

Figure 2: **DFVEdit overview.** Follow these steps for DFVEdit: (1) Encode $\mathbf{X}_0$ into the latent space $\mathbf{Z}_0$, and initialize the target latent variable as $\hat{\mathbf{Z}}_T = \mathbf{Z}_0$. (2) Transform $[\hat{\mathbf{Z}}_T; \mathbf{Z}_0]$ via the flow map $\Phi_T(\cdot)$. (3) Feed the result with prompt embeddings $[C_1, C_0]$ into the Video Diffusion Transformer, compute the delta difference to obtain the CDFV at timestep $T$, then refine it using ER and ICA. (4) Update $\hat{\mathbf{Z}}_T \to \hat{\mathbf{Z}}_{T-1}$ using the enhanced CDFV, and iterate (1)-(4) until reaching $\hat{\mathbf{Z}}_0$. (5) Decode $\hat{\mathbf{Z}}_0$ to generate the target video $\mathbf{X}_1$.

**Video editing.** Video editing via diffusion models is dominated by two paradigms: training-based and training-free methods. Training-based approaches [44–49, 12, 14] enhance pre-trained image diffusion models [21] with spatiotemporal modules, optimizing for complex edits but at high computational costs, limiting real-time applications. Conversely, training-free methods emphasize computational efficiency and real-time capability. Training-free video editing commonly involves two stages: latent space initialization and editing condition injection. Latent space initialization typically follows three paradigms: (1) forward diffusion with some steps for preserving low-frequency features [50, 51], (2) DDIM [22] inversion for enabling deterministic reconstruction [15, 17], or (3) direct source latent usage [31, 32]. For editing condition injection, most existing zero-shot methods heavily rely on *attention engineering* to maintain spatial-temporal fidelity. For instance, FateZero [15] enhances temporal consistency by caching attention maps from DDIM [22] inversion and integrating them into the denoising process; TokenFlow [17] improves spatiotemporal coherence by leveraging cached attention outputs from DDIM inversion for inter-frame correspondences and incorporating extended attention blocks during denoising; VideoDirector [20] achieves fine-grained editing via SAM [52] masks by fusing self-attention with reconstruction attention and mask guidance; and VideoGrain [19] realizes complex semantic structure modifications through SAM masks while operating on complex attention map modifications. These attention-engineered methods face scalability challenges in Transformer blocks [53], particularly for Video DiTs [2, 4] where attention memory demands grow dramatically (Fig. 1). Moreover, approaches [54–56, 24] free of attention engineering suffer from structural degradation: FRAG [54] mitigates blurring and flickering through frequency processing but compromises fidelity due to basic DDIM inversion [57] for source content retention; DMT [24] employs SSM [24] loss for motion transfer yet underperforms in detail preservation; and first-frame propagation methods (*e.g.*, StableV2V [55], AnyV2V [56]) introduce accumulating artifacts without full-frame coordination. In conclusion, designing efficient and high-quality editing methods tailored for Video DiTs remains a critical challenge.

## 3 Method

Fig. 2 provides an overview of DFVEdit. Given a source video $\mathbf{X}_0 \in \mathbb{R}^{F \times 3 \times H \times W}$ comprising $F$ RGB frames at resolution $H \times W$, together with source and target text prompts $(P_0, P_1)$, our method supports both global stylization and local modifications (shape and attribute editing). The edited

video $\mathbf{X_1}$ preserves spatiotemporal integrity in unedited regions while ensuring motion fidelity and precise alignment with $P_1$. Our approach leverages two key insights: manipulating latent space is more computationally efficient than manipulating attention (Fig. 1), and editing can be modeled as the continuous flow transformation between the source and target videos (Sec 3.1). We introduce the Conditional Delta Flow Vector (CDFV) (Sec 3.2) for this transformation. To enhance video editing performance, we utilize Implicit Cross-Attention Guidance and Embedding Enforcement (Sec 3.3) to improve spatiotemporal fidelity.

## 3.1 Unified continuous flow perspective on sampling and editing

Diffusion models include inverse and forward processes. The inverse process is typically parameterized as a Markov chain with learned Gaussian transitions, mapping noisy inputs to clean outputs. Conversely, the forward process gradually adds Gaussian noise to the clean input according to a variance schedule. As mentioned in [58, 59, 29], given a data input $x$, both inverse and forward processes can be regarded as overdamped Langevin Dynamics [60] (named Stochastic Differential Equation (SDE) in Score Matching [29]):

$$dx_t = f(x_t, t)dt + g(x_t, t)dW \tag{1}$$

where $f(x_t, t)$ is the drift coefficient corresponds to deterministic direction and $g(x_t, t)$ is the diffusion coefficient corresponds to disturbing intensity and $dW$ is a Wiener process and the probability density $P(x_t, t)$ can be described by introducing the Fokker-Planck equation [61] combined with the Ito's lemma [62] and the concept of probability flow:

$$\frac{\partial P(x_t, t)}{\partial t} = -\nabla \left[ \left( f(x_t, t) - \frac{g^2(x_t, t)}{2} \nabla log P(x_t, t) \right) P(x_t, t) \right] \tag{2}$$

Eq. 2 generalizes traditional sampling methods like DDPM [63] and DDIM [22]. ***This formulation reveals that methods based on SDE [29] obey the continuity equation principle of Flow Matching [30] and can be unified under a continuous flow perspective.*** The continuous flow is characterized by a vector field $v_t(x_t) = f(x_t, t) - \frac{g^2(x_t, t)}{2} \nabla log P(x_t, t)$, enabling state transitions from $x_t$ to $x_{t+\Delta t}$ either through flow map $\Phi_t$ in Eq. 3 or through its Euler discretized approximation in Eq. 4:

$$\begin{cases} \dfrac{d}{dt} \Phi_t(x) = v_t(\Phi_t(x)) \\ \Phi_0(x) = x \end{cases} \tag{3}$$

$$x_{t+\Delta t} = x_t + \Delta t * v_t(\Phi_t(x)) \tag{4}$$

As discussed in Section 2, zero-shot video editing includes two stages: latent space initialization and editing condition injection. The first stage involves a standard sampling process. In the second stage, we derive an isomorphism with sampling process by formulating video editing as:

$$X_{t-1}^{\text{edit}} = g_{\theta_2, t} \Big( X_t^{\text{edit}}, \underbrace{\epsilon_{\theta_1}(X_t^{\text{edit}}, t)}_{\text{Canonical Denoiser}} + \lambda \underbrace{C(X_t^{\text{edit}}, t, *)}_{\text{Control Term}} \Big) \tag{5}$$

where $\{X_t^{\text{edit}}\}_{t=0}^T$ defines the state trajectory of the edited video in the sampling process; $g_{\theta_2, t}$ is differentiable transition function parameterized by learnable $\theta_2$; $\epsilon_{\theta_1}$ is pretrained diffusion model with frozen $\theta_1$; $C(x, t, *)$ is the control term with intensity $\lambda \geq 0$ and optional extra input $*$. Under the Euler discretization scheme with step size $\Delta t \to 0$ and $\theta_2 = \mathcal{I}$, the discrete process in Eq. 5 converges to the controlled SDE:

$$dX_t^{\text{edit}} = \underbrace{\left[ -\frac{\beta(t)}{2} X_t^{\text{edit}} + \frac{\beta(t)}{2} \nabla \log p_t(X_t^{\text{edit}}) + \lambda \frac{\beta(t)}{2} \sigma(t) C(X_t^{\text{edit}}, t, *) \right]}_{f_{\theta_1}(X_t^{\text{edit}}, t)} dt + \underbrace{\sqrt{\beta(t)}}_{g(t)} dW \tag{6}$$

where $\nabla \log p_t(X_t^{\text{edit}})$ is the score function, and $\sigma(t) = \sqrt{(1 - \alpha(t))/\alpha(t)}$ is the signal-to-noise ratio coefficient with $\alpha(t) = e^{-\int_0^t \beta(s)ds}$. ***The structural isomorphism between Eq. 6 and the stochastic differential equation in Eq. 1 indicates that video editing processes can be represented within a continuous flow sampling framework***, as shown in Eq. 3 (see Appendix for more details).

## 3.2 Conditional Delta Flow Vector

Building upon the isomorphic correspondence between editing and sampling, we introduce the Conditional Delta Flow Vector (CDFV) to establish a direct continuous flow bridge from the source video to the target video.

**Delta Flow Vector.** Given the initial distribution $p(Z_T) = \mathcal{N}(Z_T; 0, I)$ for the reverse process and a clean video latent $Z$, Eq. 3 implies the existence of a time-dependent flow map $\Phi$ that:

$$Z = Z_T - \sum_{t=0}^{T} \Delta t v_t(\Phi_t(Z)) \tag{7}$$

Assuming the source and target latents $(\hat{Z}_0, Z_0)$ and their corresponding prompts $(P_1, P_0)$ are given, we replace $Z$ in Eq. 7 with $Z_0$ and $\hat{Z}_0$ respectively and define the Delta Flow Vector (DFV) as $\Delta v_t(\hat{Z}_0, Z_0) = v_t(\Phi_t(\hat{Z}_0)) - v_t(\Phi_t(Z_0))$, and the target latent $\hat{Z}_0$ can be expressed in terms of the source latent $Z_0$ as:

$$\hat{Z}_0 = Z_0 - \sum_{t=0}^{T} \Delta t \, \Delta v_t(\hat{Z}_0, Z_0). \tag{8}$$

*Eq. 8 establishes a continuous flow directly from the source latent $Z_0$ to the target latent $\hat{Z}_0$, with the vector field defined as $v_t = \Delta v_t(\hat{Z}_0, Z_0)$.* While prior works [64, 65, 31] heuristically observed that latent differences indicate editing regions, we rigorously prove this as a special case of DFV when the transformation state and vector field satisfy the continuity equation (Eq. 3).

**Conditional Delta Flow Vector.** The direct computation of $\Delta v_t(\hat{Z}_0, Z_0)$ is intractable since $\hat{Z}_0$ is the editing target. To resolve this problem, ***we leverage the terminal condition of diffusion processes to derive an unbiased estimation of DFV.*** From Eq. 2 we obtain $v_t(x_t) = f(x_t, t) - \frac{g^2(x_t,t)}{2} \nabla log P(x_t, t)$. As $t$ approaches $T$, and given that $P(x_t, t)$ is the probability density of $x_t$, if we set winner process of $Z_0$ and $\hat{Z}_0$ is equal, then $g(Z_0, t) = g(\hat{Z}_0, t)$. Consequently, as $t \to T$, both $P(Z_0, t)$ and $P(\hat{Z}_0, t)$ follow a normal distribution $\mathcal{N}(Z_T; 0, I)$ with zero mean and unit variance. Moreover, $\hat{Z}_t$ is equivalent to $Z_t$ as $t \to T$, and we have:

$$\Delta v_t(\hat{Z}_0, Z_0) \underset{t \to T}{=} f_{\theta_1, c_1}(Z_t, t) - f_{\theta_1, c_0}(Z_t, t) \tag{9}$$

The latent $\hat{Z}_{T-\Delta t}$ can be updated using Eq. 10, which corresponds to applying the continuous flow map from $\hat{Z}_0$ as defined in Eq. 11:

$$\hat{Z}_{T-\Delta t} = Z_{T-\Delta t} - \Delta t \left[ f_{\theta, c_1}(Z_T, t) - f_{\theta, c_0}(Z_T, t) \right], \tag{10}$$

$$\hat{Z}_{T-\Delta t} = \Phi_{T-\Delta t}(\hat{Z}_0). \tag{11}$$

We sequentially obtain all $v_t(\Phi(\hat{Z}_0))$ and define the Conditional Delta Flow Vector (CDFV) in Eq. 12.

$$\begin{cases} \Delta v_t(Z_0, c_0, c_1) = v_{t,c_1}(\hat{Z}_t) - v_{t,c_0}(\Phi_t(Z_0)) \\ \hat{Z}_T = \Phi_T(Z_0) \end{cases} \tag{12}$$

Theoretically, the CDFV provides an unbiased estimate of DFV. By using the CDFV as a control term, defined in Eq. 13, and integrating it into Eq. 6, we maintain a computational complexity similar to that of the basic sampling process. See the Appendix for more details.

$$C(\hat{Z}_t, t, *) = \frac{\nabla \log P(\hat{Z}_t, t) - \nabla \log P(\Phi_t(Z_0), t)}{\sigma(t)} \tag{13}$$

## 3.3 Spatiotemporal enhancement for CDFV

**Implicit Cross-Attention Guidance**. Although CDFV extracted from Video DiTs theoretically captures semantic differences between $P_0$ and $P_1$ with temporal coherence (Sec 3.2), empirical studies reveal persistent background leakage (Fig. 2). We attribute this phenomenon to the score function $\nabla_X \log p_t(X; \theta)$, which is learned by the model and may not perfectly align with theoretical

expectations. This discrepancy can introduce local distributional drift in unedited regions, and such shifts have the potential to cause noticeable alterations in the background of edited videos (see Fig. 5 for examples). Segmentation masks play a crucial role in effective structure guidance, and cross-attention, as highlighted in [16, 15, 66], exhibit significant potential for shape editing tasks. This is attributed to their time-aware adaptability and target-following characteristics, which enhance the capability to maintain structural integrity and motion consistency over time. Although most recent Video DiTs have moved from discrete cross-attention to Full Attention [1] for more accurate spatial-temporal learning, we introduce Implicit Cross-Attention derived from Full Attention. ICA still retains the essence of traditional cross-attention and guides shape editing effectively. Given text embeddings $\mathbf{E} \in \mathbb{R}^{N \times d}$ and latent video tokens $\mathbf{B} \in \mathbb{R}^{M \times d}$, Full Attention mechanism first concatenates them to form a larger matrix $\mathbf{C} = [\mathbf{E}; \mathbf{B}] \in \mathbb{R}^{(N+M) \times d}$, each row of $\mathbf{C}$ can be considered as both Query ($Q$), Key ($K$), and Value ($V$). The full attention map is computed as follows:

$$\mathcal{A} = \text{Softmax}\left(\frac{\mathbf{C}\mathbf{C}^\top}{\sqrt{d}}\right) = \begin{bmatrix} \mathcal{A}_{EE} & \mathcal{A}_{EB} \\ \mathcal{A}_{BE} & \mathcal{A}_{BB} \end{bmatrix} \in \mathbb{R}^{(N+M) \times (N+M)} \tag{14}$$

We identify that the off-diagonal block $\mathcal{A}_{EB}$ or $\mathcal{A}_{BE}$ inherently encodes cross-modal interactions. Our *Implicit Cross-Attention* extracts this block of different timesteps and binarizes it into $M_t$. We mask $\Delta v_t(Z_0, c_0, c_1)$ with $M_t$ to restrain the changes in the unedited region as Eq. 15. $M_t$ can also be optionally combined with the popular SAM [52] masks using Boolean operations.

$$\Delta v_{t,M_t}(Z_0, c_0, c_1) = M_t \odot \left[ v_{t,c_1}(\hat{Z}_t) - v_{t,c_0}(\Phi_t(Z_0)) \right] \tag{15}$$

**Target Embedding Reinforcement**. We observe that in 3D Full-Attention, the effect of text embeddings diminishes as frame length increases. This phenomenon is particularly evident in global editing tasks such as stylization. We attribute this issue to the competition between fixed-length text tokens $\mathbf{E} \in \mathbb{R}^{N \times d}$ and an increasing number of spatiotemporal tokens $\mathbf{Z} \in \mathbb{R}^{F \times H \times W \times d}$. As the video duration grows, vectors associated with stylization embeddings become increasingly sparse across frames. This sparsity may further reduce the guidance fidelity of the text embeddings. To address these challenges, we propose Embedding Reinforcement (ER) for prompt alignment:

$$\tilde{\mathbf{E}}^{(k)} = \mathbf{E} + \gamma^{(k)} \odot \mathbf{E} \tag{16}$$

where $k$ is used to locate the target embedding for editing, and its value is amplified by $\gamma + 1$. Specifically, we set $\gamma = 0.2$ for shape editing and $\gamma = 5$ for stylization. By reinforcing the embeddings, the cross-attention map is reweighted to focus on regions more relevant to the editing target, enhancing editing precision.

# 4 Experimental results

**Experimental setup.** We adopt the pretrained CovideoX-5B [1] as the base model and also extend our method to Wan2.1-14B [4] to demonstrate the robustness and flexibility of DEVEdit. All experiments are conducted on one A100-80G GPU. We evaluate our methods on public DAVIS2017 [68] videos and Internet open-source videos from Pexels [69]. In comparison experiments, we test 40-frame videos with a resolution of $512 \times 512$. Our focus is on training-free appearance editing, including local editing (shape and attribute editing), and global editing (stylization).

**Baselines.** For baselines, we compare against image diffusion-based training-free editing methods, including FateZero [15], TokenFlow [17], VideoDirector [20], and VideoGain [19], which rely on attention engineering; ControlVideo [18], FLATTEN [67], and DMT [24], which are free of attention engineering; FreeMask [16], which is based on a U-net-based video diffusion model with attention engineering; and SDEdit [50] (directly applied to CogVideoX-5B [1] base model for video editing).

## 4.1 Evaluation

**Qualitative evaluation.** Fig. 3 provides qualitative comparison results, showcasing our method's superiority in structure fidelity, motion integrity, and temporal consistency over other prominent baselines. For **single object editing** (first column), FateZero [15], TokenFlow [17], and VideoDirector [20] exhibit noticeable flickering, while ControlVideo [18], FLATTEN [67], and DMT [24]

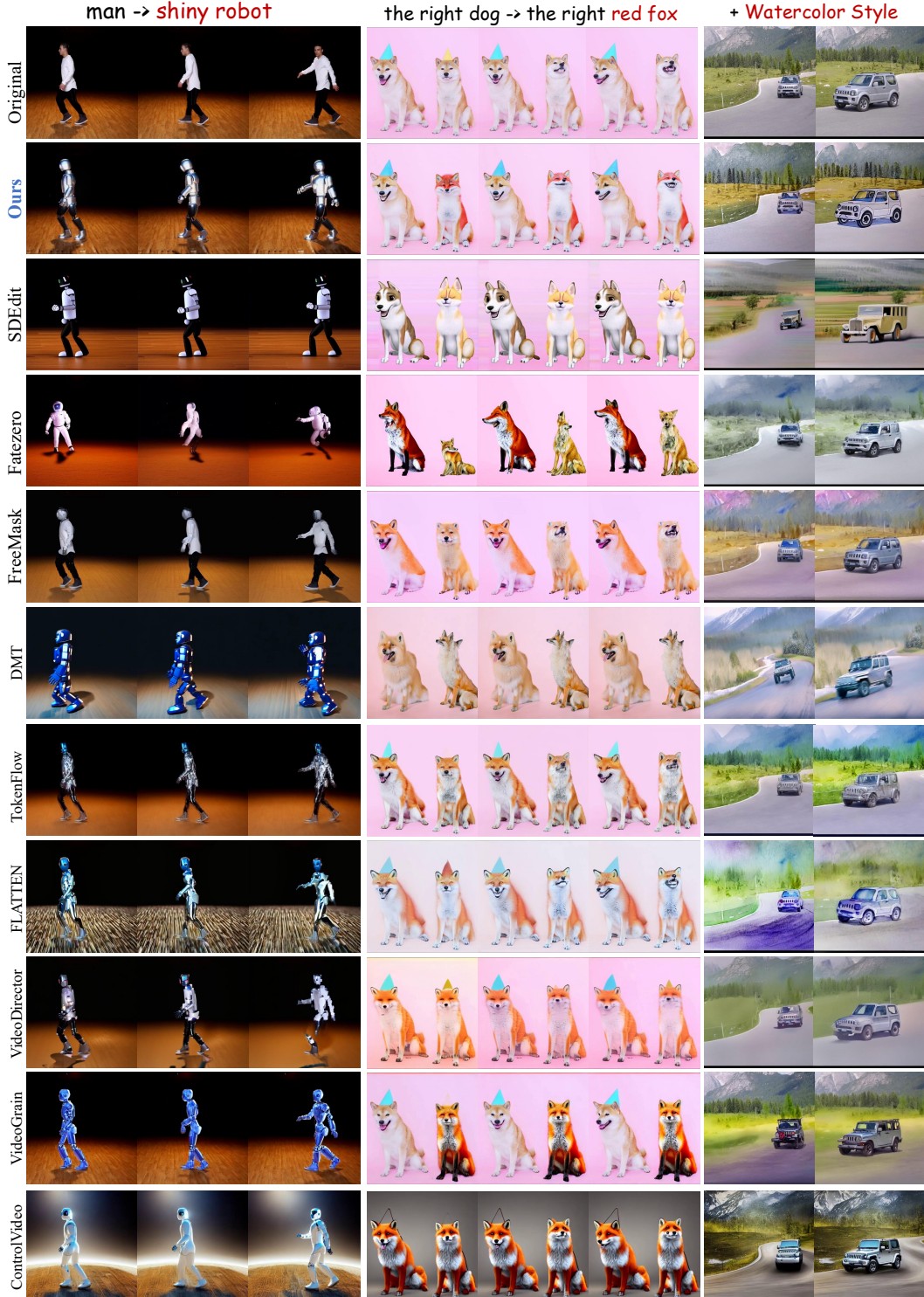

Figure 3: **Comparison.** Most methods based on attention-engineering and image diffusion models (FateZero [15], TokenFlow [17], VideoDirector [20]) suffer from flickering and fail in multi-object editing. While VideoGrain [19] enhances multi-object editing, it is inferior in structure consistency and motion detail fidelity (the second column). Attention-engineering-free approaches (FLAT-TEN [67], DMT [24], ControlVideo [18]) exhibit structural infidelity. FreeMask [16] improves temporal consistency but remains constrained by its 3D-Unet base model. Applying the image editing method SDEdit [50] directly to VideoDiTS compromises spatial-temporal fidelity. In comparison, our method achieves SOTA performance in fidelity, alignment, and temporal consistency. Refer to the supplementary material for more results.

Table 1: **Quantitative evaluation and user study results.**

| Method | Consistency | | Fidelity | | Alignment | User Study | | | Computation Efficiency | | |
|---|---|---|---|---|---|---|---|---|---|---|---|
| | CLIP-F↑ | $E_{warp}$↓ | M.PSNR↑ | LPIPS↓ | CLIP-T↑ | Edit↑ | Quality↑ | Consistency↑ | VRAM↓ | RAM↓ | Latency↓ |
| SDEdit [50] | 0.9811 | 1.67 | 20.52 | 0.4090 | 27.46 | 66.57 | 80.45 | 85.66 | 1.01 | 1.13 | **0.87** |
| FateZero [15] | 0.9289 | 3.09 | 23.39 | 0.2634 | 26.08 | 58.87 | 50.63 | 56.89 | 2.32 | 21.44 | 3.40 |
| FreeMask [16] | 0.9699 | 2.00 | 29.92 | 0.2314 | 27.06 | 75.88 | 74.67 | 77.13 | 1.64 | 25.58 | 5.65 |
| Tokenflow [17] | 0.9583 | 1.48 | 29.97 | 0.2247 | 29.78 | 70.12 | 53.45 | 57.41 | 1.43 | 3.69 | 13.03 |
| VideoDirector [20] | 0.9555 | 2.44 | 28.97 | 0.3205 | 27.50 | 74.13 | 73.25 | 71.45 | 6.00 | 2.26 | 27.97 |
| VideoGrain [19] | 0.9695 | 2.68 | 30.70 | 0.2948 | 27.79 | 76.41 | 79.87 | 70.61 | 2.35 | 2.61 | 13.44 |
| FLATTEN [67] | 0.9510 | 4.89 | 15.91 | 0.3559 | 27.57 | 63.45 | 69.45 | 68.32 | 1.54 | 7.31 | 4.61 |
| ControlVideo [18] | 0.9533 | 3.10 | 10.08 | 0.4015 | 27.06 | 56.08 | 55.33 | 59.41 | 8.74 | 1.62 | 9.45 |
| DMT [24] | 0.9668 | 3.50 | 15.95 | 0.5096 | 25.34 | 62.66 | 68.36 | 69.88 | 5.64 | 3.32 | 24.40 |
| DFVEdit | **0.9924** | **1.12** | **31.18** | **0.1886** | **30.84** | **87.65** | **84.56** | **86.98** | 0.95 | 0.86 | 1.20 |
| w/o ICA | 0.9922 | 1.25 | 29.33 | 0.1920 | 31.02 | 86.45 | 84.33 | 86.56 | 0.94 | 0.78 | 1.19 |
| w/o EmbedRF | 0.9913 | 1.13 | 31.15 | 0.1889 | 29.25 | 86.04 | 83.15 | 86.13 | 0.95 | 0.85 | 1.20 |

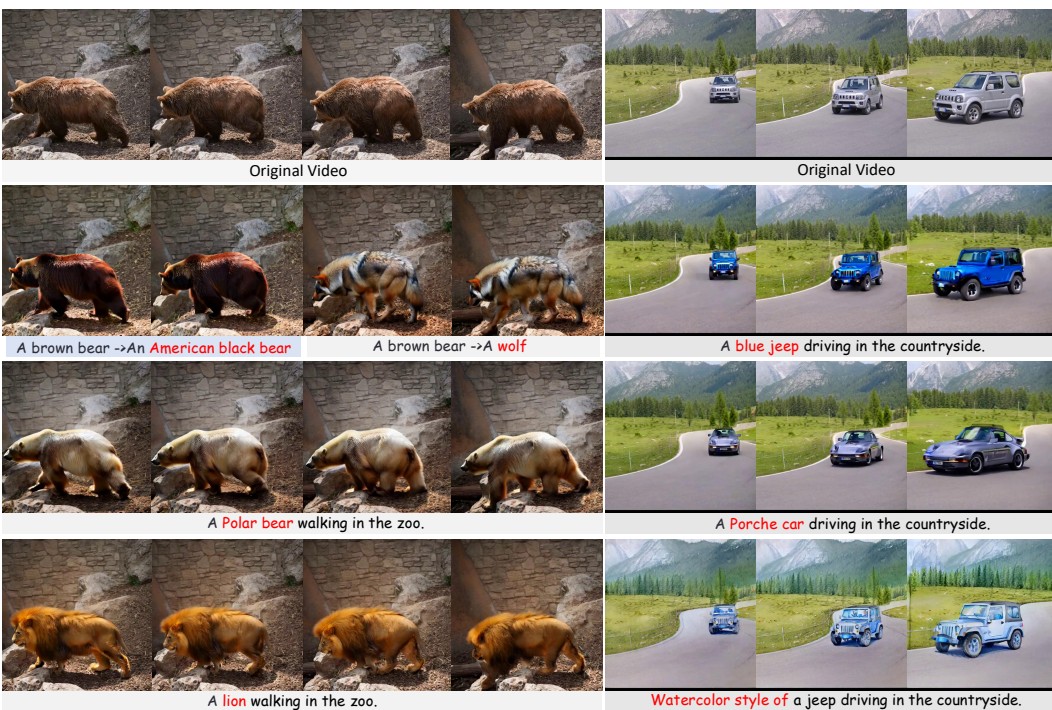

Figure 4: **Extensive qualitative results.** The extensive experiments take Wan2.1-14B [4] as the base model, demonstrating the generalization of DFVEdit for Video DiTs. See the supplementary material for more results.

fail to preserve the details of unedited regions. For **multi-object editing** (second column), most methods struggle with editing accuracy; although VideoGrain [19] achieves success in multi-object editing using fine-grained SAM [52] masks, it falls short in maintaining motion detail fidelity (e.g., a mismatch between the fox and dog expressions). For **stylization** (third column), Freemask [16], which is based on a UNet-based video diffusion model, performs notably well, while other methods still show inconsistencies in color tone and structural details (refer to the supplementary material for video displays). Additionally, we extended FateZero [15] and KVEdit [7] directly to Cogvideo-5B [1] to compare editing quality and efficiency. Due to space limitations, please refer to the appendix for more detailed comparison results. Fig. 4 provides the extensive experiment results on Wan2.1-14B [4], which also demonstrates high editing quality with respect to structure fidelity, motion integrity, and prompt alignment. Wan [4] is combined with Flow Matching [30], while Cogvideox [1] is based on Score Matching [29]. As illustrated in both Fig. 4 and Fig. 3, DFVEdit achieves consistent editing quality across popular Video DiTs, whether based on Score Matching [29] or Flow Matching [30].

**Quantitative evaluation.** In Tab. 1, we compare with baselines using both automatic metrics and human evaluations, following [15, 70, 16, 17, 12]. Specifically, **CLIP-F** calculates inter-frame cosine similarity to assess structural consistency, while $E_{\mathbf{warp}}$ measures warping error [17] to evaluate motion fidelity. Additionally, **M.PSNR** computes the Masked Peak Signal-to-Noise Ratio between

source and target videos to gauge the fidelity of unedited regions, and **LPIPS** evaluates the Learned Perceptual Image Patch Similarity for overall structural fidelity. Moreover, **CLIP-T** quantifies the alignment between the target prompt and video through the CLIP Score [71]. Regarding user studies, we focus on Target Prompt Alignment (**Edit**), Overall Editing Quality including fidelity of unedited areas, minimal filtering and blurring (**Quality**), and Motion and Structural Consistency (**Consistency**). The results demonstrate that DFVEdit achieves superior spatial-temporal consistency, fidelity, and prompt alignment compared to other methods. Furthermore, to evaluate memory and computational efficiency, we measure Relative GPU Memory Consumption (**VRAM**), defined as the ratio of editing consumption on GPU relative to original inference consumption; Relative Inference Latency (**Latency**), which assesses the ratio of editing latency to inference latency; and Relative CPU Memory Consumption (**RAM**), measuring the ratio of editing consumption on CPU over original inference consumption. These metrics highlight the practical efficiency of DFVEdit. We also extend FateZero [15] and KVEdit [7] to CogVideoX-5B [1] to evaluate their efficiencies. Some findings are illustrated in Fig. 1(b), demonstrating that these methods, originally designed for image diffusion with attention engineering, incur significant computational overhead when applied to Video DiTs.

## 4.2 Ablation results

We evaluate the efficacy of CDFV, ICA, and ER in our ablation study. In Fig. 5(a), we vary the Embedding Reinforcement factor $\gamma$ from 1 to 10. Without reinforcement ($\gamma = 1$), stylization effects are negligible. Stylization improves as $\gamma$ increases but degrades with excessively high values. Empirically, $\gamma = 5$ optimizes stylization without compromising structural fidelity or visual quality. Fig. 5(c) shows that omitting Implicit Cross-Attention Guidance leads to unintended changes in unedited regions. Incorporating cross-attention mechanisms significantly enhances structural fidelity and overall quality. In Fig. 5(b), we replace CDFV with the stochastic latent refinement vector in DDS [31]. In this ablation, for 'horse' experiment, ICA and ER are kept, while for 'bear' they are omitted for a fair comparison. The results highlights the effectiveness of CDFV. For additional qualitative and quantitative comparison and ablation results, please refer to the Appendix.

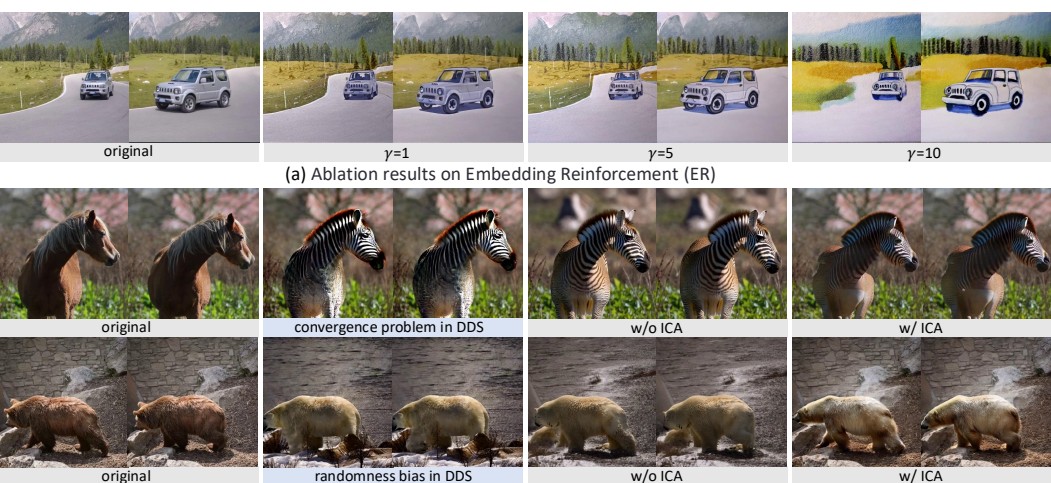

(a) Ablation results on Embedding Reinforcement (ER)

(b) Ablation results on replacing CDFV with DDS vector.    (c) Ablation results on Implicit Cross Attention (ICA) Guidance

Figure 5: **Ablation.** (a)(c) demonstrate the effectiveness of ER and ICA. (b) highlights limitations of popular approximation-based latent refinement methods [31] in video editing, including: low convergence leading to unnatural changes and unpredictable convergence times; randomness bias resulting in unsatisfactory structural fidelity. Refer to the supplementary material for more results.

## 5 Conclusion

We present DFVEdit, an efficient and effective zero-shot video editing framework tailored for Video Diffusion Transformers. DFVEdit realizes video editing through the direct flow transformation of the clean source latent. We theoretically unify editing and sampling from the continuous flow perspective, propose CDFV to estimate the flow vector from the source video to the target video, and further enhance the editing quality with ICA guidance and ER mechanism. Extensive experiments demonstrate the efficacy of DFVEdit on Video DiTs.

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
