# OpenReview forum: "DFVEdit: Conditional Delta Flow Vector for Zero-shot Video Editing"
_NeurIPS.cc/2025/Conference — Submitted to NeurIPS 2025_

### Official Review · Reviewer_EYj3 · 2025-06-24

**Clarity:** 2
**Significance:** 3
**Originality:** 4
**Rating:** 4
**Confidence:** 4

**Summary:**

This paper introduces DFVEdit, a zero-shot video editing method designed for Video Diffusion Transformers (Video DiTs). DFVEdit bypasses the need for attention modification or fine-tuning by operating directly on clean latent representations through flow transformation. The authors unify editing and sampling under a continuous flow framework and propose the Conditional Delta Flow Vector (CDFV) for estimating flow changes, enhanced by Implicit Cross Attention (ICA) and Embedding Reinforcement (ER) to improve quality. DFVEdit achieves high practical efficiency of 20× faster inference and 85% memory reduction and demonstrates state-of-the-art performance in structure fidelity, temporal consistency, and editing quality across various Video DiTs like CogVideoX and Wan2.1.

**Questions:**

[Q1] The interpretation of editing as continuous flow sampling in Line 153 is interesting. Later in Section 3.2, this correspondence is used as a basis for introducing the Conditional Delta Flow Vector. Are the formulations in Section 3.1 entirely designed by the authors?

[Q2] In Line 172, the term "unbiased estimation" is used. Could the authors clarify what is specifically meant by this in the context of their method? Furthermore, what would constitute a biased estimation in this setting?

**Ethical Concerns:**

["NO or VERY MINOR ethics concerns only"]

**Final Justification:**

I will keep my score. Thank you!

**Limitations:**

In Line 226, the paper states that its main objective is training-free appearance editing. However, it may be helpful to inform readers that other types of editing such as non-rigid modification and drag-based interaction also exist as below. While these approaches have been popular in image editing, they are increasingly being explored in video editing as well. A brief mention of these approaches could provide useful context and help position this work more clearly within the broader landscape of video editing. This could also be discussed as a potential limitation or direction for future work. Including such discussion in the related work or limitations section would strengthen the paper.

[1] DNI: Dilutional Noise Initialization for Diffusion Video Editing

[2] Drag-A-Video: Non-rigid Video Editing with Point-based Interaction

[3] DragVideo: Interactive Drag-style Video Editing

**Paper Formatting Concerns:**

The paper is well-structured and follows an appropriate format.

(This is not about formatting issue): This is a minor comment aimed at improving the paper's readability. I could not find a more appropriate section to include it, so I am noting it here.

[1] Since the caption of Figure 2 refers to components as (1), (2), and (3), it would improve clarity if the corresponding numbers were also labeled directly in the figure to facilitate easier cross-referencing.

[2] In Line 217, the term "CovideoX-5B" appears to be a typo and should be corrected.

**Quality:**

3

**Strengths And Weaknesses:**

**Strengths**

[1] The paper correctly identifies the limitations of existing heuristic methods that require attention manipulation for diffusion-based video editing and presents a compelling alternative that achieves strong results.

[2] The theoretical formulation that interprets editing as a form of sampling from a distribution flow perspective is well-grounded. I consider this to be the core contribution of the paper.

[3] The visual quality of the generated videos is strong, and the authors provide a wide range of comparisons, which enhances the credibility of their results.

**Weakness**

[1] Since the Conditional Delta Flow Vector (CDFV) is designed to operate under a large number of diffusion steps T, it would be beneficial to provide ablation results showing how output quality varies with different values of T.

[2] In some qualitative results, such as the "moonwalking" video, the baseline method (e.g., VideoGrain) appears to produce higher visual fidelity, especially in rendering the floor, and shows better textual alignment with the "red fox" edited video. But, why is there such a noticeable gap in quantitative performance compared to the proposed method?

[3] While the method avoids explicit attention modification, the Implicit Cross-Attention (ICA) mechanism could still be influenced by masking and editing quality. If so, could the authors clarify the practical advantage of ICA over attention modification methods that do not rely on masks? A comparison or discussion in this regard would help clarify the benefits of the proposed approach.

---

> ### Author Rebuttal · Authors · 2025-07-30
>
> We extend our sincere gratitude for your visionary assessment, which recognized the theoretical novelty in our unification of editing and sampling. Your constructive suggestions on ablation studies helped crystallize the mathematical essence of our framework. Below are our responses:
>
> ### Weaknesses
>
> #### **1. On Ablation Study of Number of Diffusion Steps $T$**
>
> We thank the reviewer for this suggestion. To analyze the sensitivity of DFVEdit to the total number of diffusion steps (T), we conducted an ablation study on Wan2.1-14B with $ T ∈ \{10, 20, 50, 100, 150, 200 \}$. As shown in **Table R5**, both editing accuracy (measured by CLIP-T↑) and temporal consistency (E_warp↓, CLIP-F↑) stabilize when T≥50. For T < 50, insufficient flow field estimation leads to under-editing, manifesting as blurred or temporally unstable results. In contrast, increasing *T* beyond 50 yields diminishing improvements at the cost of higher computational overhead, and may result in a slight decrease in performance on background preservation. We therefore set $T=50$ as the default, achieving an optimal trade-off between efficiency and quality. This value is also consistent with common practice in high-fidelity video generation, where moderate step counts suffice under advanced solvers. The quantitative and qualitative visualization results will be included in the final version.
>
>
>
> *Table R5 Ablation Study on Number of Diffusion Steps $T$*
>
> | Total Timesteps *T* | CLIP-F↑    | E_warp↓  | M.PSNR↑   | LPIPS↓     | CLIP-T↑  |
> | ------------------- | ---------- | -------- | --------- | ---------- | -------- |
> | 10                  | 0.9731     | 2.1305   | 30.56     | 0.1793     | 29.91    |
> | 20                  | 0.9833     | 2.11     | 30.61     | 0.1688     | 30.66    |
> | 50 (default)        | 0.9950     | **1.06** | **31.23** | **0.1568** | 31.34    |
> | 100                 | 0.9954     | 1.4175   | 30.96     | 0.1569     | 31.32    |
> | 150                 | 0.9973     | 1.3881   | 30.73     | 0.1571     | 31.36    |
> | 200                 | **0.9953** | 1.3672   | 30.66     | 0.1578     | **31.5** |
>
>
>
>
>
> #### **2. On Qualitative–Quantitative Discrepancy**
>
> We acknowledge that VideoGrain has good performance on static details (e.g., floor textures) due to its use of fine-grained SAM masks for local optimization. However, this does not imply superior video editing performance. In dynamic consistency—our key focus—DFVEdit outperforms VideoGrain. For example, in the case of the red fox (Figure 4), VideoGrain alters motion pattern details, resulting in unnatural movement (see supplementary video), despite achieving good text alignment. And on stylization tasks ( without SAM masks), VideoGrain's structure preservation ability decreases. In contrast, DFVEdit preserves original motion dynamics while achieving accurate appearance editing, which is supported by lower temporal distortion (E_warp↓) and higher user scores in spatial-temporal consistency (Consistency↑).
>
>
>
> #### **3. On ICA vs. Traditional Attention Engineering**
>
> We thank the reviewer for this question. It is essential to clarify that **ICA is not an attention engineering component**, nor does it modify attention weights or feature activations. In contrast to attention-engineering-based approaches that directly manipulate keys, values, or attention maps, **ICA only infers an implicit spatial mask** from attention responses to guide the *Conditional Delta Flow Vector (CDFV)*—our core editing mechanism.
>
> This fundamental distinction brings three key advantages:
>
> - *Preservation of feature distribution*: ICA does not alter attention outputs or introduce residual perturbations, avoiding unintended distribution shifts in unedited regions.
> - *Computational efficiency*: The mask is computed once per timestep via a forward pass. In contrast, attention engineering requires iterative recomputation and caching across layers.
> - *Architecture compatibility*: As a lightweight plugin to CDFV, ICA does not rely on specific attention layouts (can be extracted from both cross-attention in U-Net and full-attention in DiT), enabling seamless deployment on modern Video DiT models where traditional attention editing faces integration and scalability challenges.
>
> In essence, ICA enhances editing accuracy without engaging in attention feature manipulation, making it fundamentally different from and more scalable than conventional attention engineering.
>
>
>
> ### Questions
>
> #### **Q1. On the Originality of Section 3.1**
>
> We sincerely thank the reviewer for this thoughtful question. We would like to clarify the motivation and contribution of Section 3.1. The proposed framework, which unifies sampling and editing through a continuous flow perspective, is inspired by the theoretical foundations of Score Matching [1] and Flow Matching [2], particularly the continuous flow interpretation of SDEs discussed in prior work (e.g., in the appendix of the Flow Matching literature). We adapt this idea to the video editing task, offering a new viewpoint that models editing as a latent space flow transformation. Equations 1–4 reframe Score Matching and Flow Matching within a unified continuous flow formulation, providing a clearer theoretical foundation for editing design. Furthermore, Equations 5–6 represent our *original derivation*, which unifies sampling and editing, explicitly connects flow prediction to practical editing operations, and directly motivates the design of the CDFV. While built upon prior theory, the extension offers a principled and insightful step toward more coherent and controllable video editing.
>
> ####  **Q2. On the Term "Unbiased Estimation"**
>
> We thank the reviewer for this clarifying question. In our context, **"unbiased estimation"** means that the expected value of the Conditional Delta Flow Vector (CDFV) equals the true theoretical Delta Flow Vector (DFV) $\Delta v_t$ (as defined in L164). This ensures the CDFV accurately captures the intended conditional flow shift from source to target latents. The unbiasedness of CDFV relies on three key conditions: terminal distribution convergence, shared (coherent) noise, and fine discretization (discussed in Section 3, L170–176).
>
> A **biased estimation** arises when these conditions are violated, such as:
>
> * *Excessive edit strength:* When $|\hat{Z}_0 - Z_0| > 3\sigma$, the terminal latent $\hat{Z}_t$ may deviate from the standard Gaussian prior $\mathcal{N}(0, I)$ at $t=T$, breaking the terminal convergence assumption. This aligns with our stated limitation regarding extreme geometric deformations.
>
> * *Non-shared noise:* If the source and target paths use independent noise realizations ($dW_{\hat{Z}} \ne dW_Z$), the stochastic components do not cancel, introducing noise-induced bias into CDFV.
>
> * *Large $\Delta t$:*  The unbiasedness holds strictly in the continuous-time limit ($\Delta t \to 0$). Large step sizes introduce discretization error, leading to estimation bias.
>
>
>
>
>
> ### Limitations
>
> We sincerely thank the reviewer for the valuable feedback. In the final version of the paper, we will add a dedicated discussion in the Limitations section to clarify that DFVEdit is primarily designed for *appearance editing*, and currently has limited support for non-rigid shape modifications or drag-based interactive editing. We will also cite and discuss the relevant works recommended by the reviewer [1–3] in both the Related Work and Limitations sections to strengthen the paper.
>
>
>
> ### Paper Formatting Concerns
>
>  We thank the reviewer for the careful reading. We will revise Fig. 2 in the final version to include explicit annotations (1)–(5) corresponding to the key components in the pipeline. Additionally, the typo at Line 217—'CovideoX-5B'—has been corrected to 'CogVideoX-5B'.
>
>
>
> [1] Song, Y., Sohl-Dickstein, J., Kingma, D. P., Kumar, A., Ermon, S., & Poole, B. (2020). Score-based generative modeling through stochastic differential equations.
>
> [2] Lipman, Y., Chen, R. T., Ben-Hamu, H., Nickel, M., & Le, M. (2022). Flow matching for generative modeling

---

> > ### Comment · Reviewer_EYj3 · 2025-08-05
> >
> > For the response to weakness 2, [1] You also mention that VideoGrain maintains high fidelity thanks to SAM masks, but aren't your own methods also using SAM masks? [2] Isn't your method also relying on SAM masks? it's a bit hard to understand why you're emphasizing the gain without using masks.

---

> > > ### Author Response · Authors · 2025-08-05
> > > **Clarification on the Role of SAM in DFVEdit**
> > >
> > > Thank you for pressing us to clarify this fundamental distinction. We apologize for any prior ambiguity. To directly address your concern:
> > >
> > > Does DFVEdit use SAM masks? Yes, but only as an optional enhancer in multi-object editing (as noted in Appendix L106–L114). Crucially, **the critical gain of DFVEdit comes not from eliminating SAM masks** but from a **fundamentally different and more flexible use of masks**. Below are key differences:
> > >
> > > **1) Mask Dependency**
> > >
> > > - **VideoGrain** is an outstanding work that achieves class-level, instance-level, and part-level edits. It requires precise SAM masks as essential input for attention editing and fails without them.
> > > - **DFVEdit** operates **by default without SAM**. Spatial constraints come from ICA, which produces *timestep-aware, coarse-to-fine* masks (see Figure 5, F2). **SAM masks are only optionally used in multi-object editing** (e.g., “the right dog -> the right fox” in Figure 3  ) to aid ICA mask localization, considering that pure ICA can suffer from ambiguous spatial binding due to shared semantic features in multi-objects (a limitation also noted in works like FreeMask). **No SAM masks are used in single-object editing and stylization** for DFVEdit, yet DFVEdit outperforms VideoGrain (see Figure 3, columns 1 & 3; supplementary video, 0.5× speed recommended—e.g., ‘man → robot’ shows sharper temporal coherence and smoother shape transitions in DFVEdit, while VideoGrain exhibits jagged edges and inconsistent detail preservation.
> > >
> > > **2) Mask Processing**
> > >
> > > - We do not use raw SAM masks. Instead, we apply **padding** to soften edges (as region prior, not strict boundary), and take the **intersection with ICA maps** to get timestep-aware, coarse-to-fine masks. The result is that masks remain **soft and diffusion-aligned**, avoiding the rigidity of SAM. As noted in Appendix L114, we may use the padded SAM mask alone in later denoising steps to prevent over-constraining from fine-grained ICA–SAM intersections, further enhancing editing fluidity.
> > >
> > > **3) Mask Application**
> > >
> > > - **VideoGrain** applies SAM masks in attention layers (modifies Q/K/V), a classical attention-engineering-based method that may incur substantial computational overhead when directly applied to Video DITs.
> > > - **DFVEdit** applies masks to **constrain the CDFV in the latent space**. This preserves attention dynamics, enables end-to-end deformation, and supports model-agnostic deployment (validated on U-Net and various Video DiTs).
> > >
> > > **4) The purpose of SAM in our work** mainly addresses semantic leakage of cross-attention in multi-object cases (e.g., ICA confusing left/right dog). SAM provides coarse location; ICA provides temporal evolution. SAM guides ICA, not replaces it. We apologize for the omission of this clarification and thank the reviewer for prompting it.
> > >
> > > Therefore,  DFVEdit reduces reliance on external masks by design. When SAM is used, it is softened and fused with ICA, enabling more natural, robust editing. The gain lies in *how* we use masks, not in *not using* them. We will clarify this in the final version.

---

### Official Review · Reviewer_Q6ss · 2025-07-01

**Clarity:** 1
**Significance:** 2
**Originality:** 3
**Rating:** 3
**Confidence:** 3

**Summary:**

The authors propose Conditional Delta Flow Vector (CDFV) for video editing based on Video DiTs. Instead of using attention-engineering or further finetuning, CDFV operates directly on the input latents, achieving 20x faster inference speed and 85% less memory cost. CDFV achieves a SoTA performance as a zero-shot video editing method for video DiTs.

**Questions:**

- How can you figure out which token is for stylization? (L209 and Fig. 2) By a manual segmentation?
- This is a little minor but will the proposed methods work for other aspect ratios instead of only for squared (1:1) ones?

**Ethical Concerns:**

["NO or VERY MINOR ethics concerns only"]

**Limitations:**

The authors have included the limitations of the approach, including the preservation the detail fidelity and editing with large variations. However, it is not clear if the failure of preservation comes from the proposed approach or from the compression loss in the VAE.

**Paper Formatting Concerns:**

No concerns.

**Quality:**

2

**Strengths And Weaknesses:**

The paper tries to resolve a critical issue for video editing in DiTs for memory costs. It works on latent inputs instead of attention maps. The proposed method is zero-shot and training-free. The quality of the results is acceptable. My major concerns include: (a) It turns out in practice the authors still use attention maps and do attention engineering (ICA), (b) The writing of the paper is not well-organized, especially in Fig. 1 and Fig. 2, and (c) The examples shown in the supplementary material are relatively easy, with small motions and dynamics.

My current rating for the paper is Borderline Reject.

## Strength
### Idea/Motivation
- The authors analyze why previous attention-engineering methods and finetuning methods have difficulties in video DiTs in Fig. 1. This provides a useful evidence for introducing a new method with less speed and memory overheads.
- The authors operate directly on the latent inputs which makes sense in the context of video DiTs then.

### Method
- The authors start with DFV and CDFV with a theoretical explanation, introducing CDFV for an unbias estimate for DFV.
- To mitigate the discrepancy between the theoretical basis and the empirical shift, the authors introduce ICA and Target Embedding Reinforcement (ER).

### Experiments/results
- The authors conduct a comprehensive comparison with previous methods in Table 1 and Fig.4, showing that DFVEdit outperforms these methods in a zero-shot fashion.
- The ablation study on ER factor $\gamma$ is interesting.
- With CDFV, the temporal consistency improves a lot in the video results.

## Weaknesses
### Method
- It is not clear to me why the proposed method is different from previous attention-engineering works conceptually because ICA and SAM (optionally though) are used.

### Experiments
- Only easy cases are shown in the paper. Does DFVEdit work for videos with larger dynamics and motions?
- In the Zebra case in the supplementary material, it is not clear to me what the benefit of ICA is as I personally prefer the result w/o ICA. In the result with all components, the mane of the horse looks much unnatural, although the preservation of the background is achieved.

### Writing/Delivery
- Fig. 1(c) is hard for readers to understand in my opinion. Without any further explanations, readers do not understand what the meaning of "Normalized DFV" is and how this figure is related to the topic. It is highly recommended the authors include explanations either in the caption or in the main text, given Fig. 1 plays a critical role demonstrating the motivation of the problem.
- The name "DFVEdit" comes out of nowhere in the Introduction section (L69). It needs a definition for its debut.
- Fig. 2 lacks a clear guidance between the figure and the caption. For example, what are (1), (2), (3), (4) and (5) in the caption and how can readers associate them in the figure?
- Also the notations in Fig.2 are messy. (a) Sometimes the authors use capitalized "T" but sometimes it is lower-case "t". (b) $\gamma$ is not defined at all in "Embedding Reinforcement" in this figure. It is suggested all symbols are self-contained in a figure. (c) What is $\otimes$ in Embedding Reinforcement?
- Overall, Fig. 2 is not well-organized and does not help readers understand the proposed method.

---

> ### Author Rebuttal · Authors · 2025-07-30
>
> We sincerely appreciate your precision-driven review, which highlighted ambiguities in the presentation. Your laser-like focus on clarity – from figure annotations to terminology definitions – has significantly enhanced the paper's technical communicability. Below are our responses:
>
> ### Weaknesses
>
> ### Method
>
> **1. About the difference from previous attention-engineering works.**
>
> * **Difference on the conceptual foundation**: We unified editing and sampling, and model editing as a *continuous flow transformation* in latent space via CDFV (Eq.8-12), whereas attention engineering *heuristically manipulates feature correlations* via Q/K/V modifications.
> * **Difference on the operational mechanism**: CDFV performs end-to-end deformation and ICA only *infers an implicit mask* from cross-attention or full-attention maps to constrain CDFV’s spatial influence and SAM masks are also applied to CDFV optionally (used in multi-object shape editing), which means ICA and SAM masks are operated on the latent space without engaging in traditional attention engineering.
> * **Difference on the architectural impact**: DFVEdit's flow-based formulation enables **model-agnostic operation** (have been verified on 2D U-Net in Table R1 and various Video DiTs), while most attention engineering works fail on Video DiTs due to architectural and computational constraints
>
>
>
>
>
> ### Experiments
>
> **1. On Case Complexity**
>
> We appreciate the reviewer’s comment. Our evaluation covers diverse motion regimes. Due to the limitations of static figures, dynamic behaviors may not be fully conveyed. We invite the reviewer to consult the supplementary video, which clearly illustrates temporal coherence, editing fidelity, and motion dynamics. To evaluate motion preservation under diverse conditions, we designed experiments across four categories of motion complexity and articulation level:
>
> - *High Dynamics, Low Articulation*: e.g., *jeep stylization* – large motion with minimal structural change.
> - *Moderate Dynamics, Moderate Articulation*: e.g., *man → robot* – involving limb articulation and pose transitions.
> - *Low Dynamics, High Articulation*: e.g., *dogs → foxes* – preserving fine, high-frequency non-rigid motions (ear flicks, blinking, facial expressions).
> - *High Dynamics, High Articulation*: e.g., *cyclist editing* – combining fast motion, limb articulation, and micro-action transfer (see Extensive Results III in the supplementary video).
>
> DFVEdit preserves spatio-temporal coherence from global trajectories to local pose details. In addition, thanks to its resource-efficient design, DFVEdit enables long-sequence editing (beyond typical 8–20 frame limits), enhancing motion continuity. We will include more long-duration, complex cases in the final version.
>
> **2. On ICA’s Effect in the Zebra Case**
>
> We thank the reviewer for the insightful observation. The zebra case illustrates a fundamental trade-off in ICA: without ICA, higher edit intensity (e.g., mane transformation) but severe background degradation; with ICA, Strong background preservation, yet potentially conservative editing on *dynamic, ambiguous boundaries* (e.g., flowing mane). This stems from current limitations in modeling diffuse edges under motion. Nevertheless, quantitative results (Table 1) show consistent gains in ‘M.PSNR’ and ‘LPIPS’, and qualitative examples (e.g., bear→polar bear in Figure 5, multi-object edit in Figure F3) confirm ICA’s critical role in visual fidelity and editing accuracy. The benefits of background protection outweigh the minor loss in edge edit strength in most cases. We will explicitly discuss this limitation and explore adaptive masking mechanisms in future work.
>
>
>
>
>
> ### Writing/Delivery
>
> We appreciate the reviewer's detailed feedback on the paper's clarity and readability. Below are the revisions we will implement in the final version:
>
> **(1) Caption and Text Supplement for Fig 1(c)**: "Normalized DFV" refers to the original DFV normalized using Min-Max scaling to the range [0,1], illustrating its relative intensity distribution across different sampling steps *t* and spatial positions. This figure reveals the evolution pattern of DFV from **coarse contours to fine details**, consistent with the diffusion sampling process, providing an intuitive motivation for a unified perspective on editing and generation.
>
> **(2) Definition of Terms at First Mention**: We will add the full name and a brief explanation of "DFVEdit" at its first mention in the Introduction (line 69), enhancing readability.
>
> **(3) Redrawing of Figure 2 Overview**: Given that the current Figure 2 is not conducive to understanding, we will thoroughly redraw it after rebuttal, emphasizing five key stages and ensuring all module locations and data flows are clearly labeled to enhance interpretability. The improved description is as follows:
>
> 1. **Latent Initialization**: Starting from the far left of the diagram, marked as (1), it clearly illustrates the process of obtaining noise-free initial latent variables from input videos through encoding.
> 2. **Forward Flow Mapping**: In the middle section of the diagram, marked as (2), it shows how image features are refined through a series of flow transformations from the latent variables in the previous step.
> 3. **Condition Embedding Feeding & CDFV Computation**: Parallel to the forward flow and positioned towards the upper right, marked as (3), it highlights how text condition is embedded into the model and the specific process of CDFV computation based on these conditions.
> 4. **Latent Update**: Returning to the top left of the diagram, marked as (4), it explains how latent variables are updated based on new information after condition embedding and CDFV computation.
> 5. **Iterative Refinement → Output**: At the bottom left of the diagram's first row, marked as (5), it depicts the final step—iterative refinement until generating high-quality output images.
>
> **(4) Symbol Consistency and Mechanism Clarification**: *t*: Current diffusion time step; *T*: Total number of sampling steps (constant). In the ER module, *γ* represents the reweighting strength (see L205–215). The ER sub-figure shows: input text embeddings → key token identification (red box) → reweighting process (pink box on the right) → influence on Key/Value in ICA (left side), and a comparison of ICA responses before and after embedding enhancement (right side), visually explaining its mechanism.
>
>
>
> ### Questions
>
> **Q1.How can you figure out which token is for stylization? (L209 and Fig. 2) By a manual segmentation?**
>
> Our token recognition process is fully automated, and it proceeds as follows:
>
> Step 1: Syntax Marking - Users add special delimiters `[]` around target concepts (such as style words) in their editing instructions, for example, `"[Watercolor style painting] of a castle at sunset"`.
>
> Step 2: Token Mapping and Localization - The text is converted into a token sequence using a tokenizer. Simultaneously, the tokenizer maps the special delimiters to a token sequence, which serves as a matching substring to locate the positions of these delimiters within the token sequence, thereby automatically identifying the corresponding tokens. Once identified, the weights of these corresponding tokens are adjusted through reweighting (Eq.16).
>
>
>
> **Q2. This is a little minor but will the proposed methods work for other aspect ratios instead of only for squared (1:1) ones?**
>
> Regarding aspect ratio support, DFVEdit fully supports video input with any aspect ratio. The method itself does not have a square constraint on the spatial dimension of the input latent variables. We used squares in the experiment only for the consideration of uniform preprocessing and common formats. The final version will include examples of editing results for videos with non-1:1 aspect ratios (such as 16:9).
>
>
>
> ### Limitations
>
> We agree that the reasons for editing failures (such as insufficient fidelity of background details) are partly due to compression loss in the VAE, which is a common bottleneck in current latent space diffusion models, especially affecting the restoration of high-frequency textures and delicate structures. However, we also note that the design limitations of the method itself also play an important role. For example, in Appendix Figure F4, "chimpanzee → chimpanzee with sunglasses" can better preserve the details of the background banner, while "chimpanzee → chimpanzee with red hat" shows texture degradation of the background banner.

---

### Official Review · Reviewer_XgWS · 2025-07-02

**Clarity:** 2
**Significance:** 3
**Originality:** 3
**Rating:** 4
**Confidence:** 3

**Summary:**

This paper proposes DFVEdit, an efficient zero-shot video editing method that operates through continuous flow transformations in a clean latent space, introducing Conditional Delta Flow Vector (CDFV) combined with Implicit Cross-Attention (ICA) and Embedding Reinforcement (ER) to enhance editing performance. Experiments demonstrate effectiveness.

**Questions:**

As detailed in the weakness section, my primary concerns focus on the zero-shot design motivation, the overly complex method architecture, limitations in the experimental results, and the fairness of experimental comparisons.

**Ethical Concerns:**

["NO or VERY MINOR ethics concerns only"]

**Final Justification:**

My concerns regarding method design, experimental results, and comparison fairness have been well addressed. While I still have some reservations about the upper bound of zero-shot video editing in this era, the new perspective of unifying editing and sampling under continuous flow is valuable.

**Limitations:**

Yes, discussed in the Supplementary material.

**Quality:**

3

**Strengths And Weaknesses:**

Strengths
1. The overall writing logic of the paper is clear.
2. This paper theoretically unifies generation and editing through derivation, proposing CDFV, Embedding Reinforcement, and Implicit Cross-Attention Guidance to achieve zero-shot video editing, obtaining performance superior to other methods.

Weaknesses
1. Concerns about zero-shot design. First, from a user perspective, they don't care about training resource consumption; their primary concerns are editing performance and quality, followed by inference latency. Moreover, zero-shot methods often have significantly inferior results compared to fine-tuning methods. Therefore, such methods need to provide more insights to help researchers and users understand the essence of video editing or assist in designing better training-based methods. Training-based methods only require some resource consumption during training, but the resources needed during inference after training are comparable. If the motivation is merely mentioning that training methods require some computational resources for fine-tuning, this motivation is somewhat unconvincing.
2. Bloated method design. The framework contains quite a few small components. Besides the proposed CDFV, it also includes Embedding Reinforcement and Implicit Cross-Attention Guidance. Such complex design doesn't result in particularly good final performance, making the entire framework appear somewhat bloated with many parameters to tune.
3. Shortcomings in experimental results. In terms of capabilities, the results shown in the paper, including the mp4 files in supplementary materials, mainly focus on style transformations, such as from "bear" to "polar bear," "natural scenery" to "ink painting" style. I'm curious whether the method doesn't support editing of shapes, motion patterns, or actions. If so, what are possible solutions? Or said, does it also demonstrate the limited upper bound of zero-shot methods?
4. Fairness of experimental comparisons. Other compared methods, such as FateZero, TokenFlow, VideoDirector, ControlVideo, etc., are mainly based on 2D base models like Stable Diffusion, so it's natural for them to show some deficiencies in temporal consistency. From a fairness perspective, shouldn't this paper also conduct experiments based on Stable Diffusion 1.5's image model to compare with other methods?

---

> ### Author Rebuttal · Authors · 2025-07-29
>
> We are grateful for your incisive conceptual critique that challenged us to refine the philosophical grounding of DFVEdit. Your sharp perspective on zero-shot motivation prompted us to articulate broader value propositions that extend beyond efficiency gains. Below are our responses:
>
>
>
> ###  Weaknesses:
>
> #### **1. On Zero-Shot Design and Practical Value**
>
> * The strength of zero-shot methods lies not only in eliminating the need for training but also in their *plug-and-play flexibility* and *model-agnostic applicability*. This is especially valuable for non-expert users and real-world applications, where retraining for new tasks or a new model (e.g., rapidly evolving Video DiTs) is impractical due to closed-source models or high computational costs.
> *  DFVEdit offers a unified perspective on generation and editing through continuous manifold transformations in latent space, and proposed CDFV reveals that editing can be interpreted as a conditional flow field shift, providing theoretical insight beyond mere performance.
> *  While fine-tuning methods (e.g., Tune-A-Video[2], VideoP2P[3]) may achieve high performance on specific tasks, DFVEdit delivers *strong editing quality with zero-shot convenience*, matching or even exceeding them in temporal consistency and background preservation, as validated in comparison experiments with the fine-tuning method DMT[3] (see Figure 7, Table 1). To further validate this, we include more comparison results with fine-tuning baselines in **Table R4**.
>
>
>
> *Table R4: Comparison with fine-tuning-based methods.*
>
> | Method                      | CLIP-F↑ | E_warp↓ | M.PSNR↑ | LPIPS↓ | CLIP-T↑ |
> | --------------------------- | ------- | ------- | ------- | ------ | ------- |
> | VideoP2P [1]                | 0.9624  | 3.28    | 17.38   | 0.4531 | 27.26   |
> | Tune-A-Video [2]            | 0.9612  | 3.35    | 16.67   | 0.4545 | 27.14   |
> | DMT [3]                     | 0.9668  | 3.50    | 15.95   | 0.5096 | 25.34   |
> | **DFVEdit** (CogVideoX-5B)  | 0.9924  | 1.12    | 31.18   | 0.1886 | 30.84   |
> | **DFVEdit** (on Wan2.1-14B) | 0.9950  | 1.06    | 31.23   | 0.1568 | 31.34   |
>
>
>
>
>
> #### **2. On Method Design Simplicity and Necessity**
>
> We thank the reviewer for bringing this vital point to our attention. While simplicity is desirable, the three core components of DFVEdit—CDFV, ICA, and ER—are distinct and indispensable.
>
> - **CDFV** establishes the theoretical foundation, modeling editing as flow field transformation in latent space—defining *how* to edit, and is the core contribution of our method.
> - **ICA** prevents "local distribution drift" by implicitly constraining latent updates to target regions, preserving background and temporal consistency—ensuring *what not to change*, is a protection module for CDFV.
> - **ER** enhances prompt alignment by adaptively reweighting key tokens in the text prompt, ensuring *what to edit* aligns with intent.
>
> Ablation studies confirm that removing any component leads to significant degradation: ICA removal causes background artifacts; ER removal reduces prompt alignment and CLIP-T scores. Visual comparisons in the Appendix further validate their necessity. The framework introduces easy-tunable hyperparameters (ICA mask fusion timesteps, ER reweighting strength), which exhibit robustness across tasks. This demonstrates its practicality. In summary, DFVEdit’s design is function-driven: each component solves a specific, empirically validated challenge. The architecture reflects engineering clarity, balancing editing strength, consistency, and instruction alignment, without redundancy.
>
>
>
> #### **3. On Experimental Scope and Dynamic Editing**
>
> We appreciate the reviewer’s feedback on case diversity. DFVEdit supports not only style transfer but also:
>
> * Single object shape editing (see Figure 3,' man -> robot ';  Figure 4)
>
> * Object addition/removal and attribute editing (Figure F4)
>
> * Multi-object editing (Figure 3, 'the right dog -> the right fox';  Figure F3)
>
> Due to the limitations of static figures, dynamic behaviors may not be fully conveyed. **We invite the reviewer to consult the supplementary videos**, which clearly illustrate temporal coherence and editing fidelity. DFVEdit maintains *strong temporal consistency* across all categories, from global trajectories to local pose details, outperforming existing zero-shot methods and, in several metrics, rivaling fine-tuned approaches (e.g., DMT, VideoP2P, Tune-A-Video, as shown in Table R4).
>
> We clarify that this work focuses on appearance editing (as stated on p.6, line 221), not motion editing (e.g., changing "walking" to "dancing"). Such tasks require explicit motion control (e.g., via optical flow, 3D poses), and remain a key direction for future work.
>
>
>
> #### **4. On Fairness of Experimental Comparisons**
>
> We appreciate the reviewers’ insightful comments regarding experimental fairness across model architectures. Indeed, direct comparisons between methods built on different backbones, we rigorously address fairness concerns through:
>
> * In comparison, we have adopted SDEdit (as shown in Figure 3 and Table 1) on the base model CogVideoX-5B, as well as the CogVideoX-V2V pipeline (presented in Table R2), for a fair comparison, which demonstrates that even on the same base model, DFVEdit still outperforms.
>
> * We have conducted controlled experiments using a unified backbone: Stable Diffusion v1.5 (2D U-Net). In this setting, we evaluate DFVEdit’s core components (CDFV, ICA, and ER) in image editing tasks (as shown in Table R1), where the temporal modeling advantages are removed. Results show that DFVEdit significantly improves editing accuracy, detail preservation, and semantic alignment, demonstrating its general effectiveness independent of architectural strength.
>
> To the best of our knowledge, we are the first work enabling effective and efficient zero-shot video editing on **modern Video DiT architectures** — including both *score-matching-based* (e.g., CogVideoX) and *flow-matching-based* (e.g., Wan2.1) formulations. Prior attention-based editing methods are often incompatible with DiTs due to architectural constraints and high computational cost. DFVEdit overcomes these limitations through a lightweight, flow-field-based formulation.
>
> Thus, while comparisons across architectures require careful interpretation, our results highlight two complementary strengths:
>
> * **General applicability**—validated under controlled 2D settings;
> * **Forward-looking capability**—unlocking high-quality, zero-shot editing on powerful Video DiT models that prior methods cannot support.
>
>
>
> [1] Liu, S., Zhang, Y., Li, W., Lin, Z., & Jia, J. (2024). Video-p2p: Video editing with cross-attention control.
>
> [2] Wu, J. Z., Ge, Y., Wang, X., Lei, S. W., Gu, Y., Shi, Y., ... & Shou, M. Z. (2023). Tune-a-video: One-shot tuning of image diffusion models for text-to-video generation.
>
> [3] Yatim, D., Fridman, R., Bar-Tal, O., Kasten, Y., & Dekel, T. (2024). Space-time diffusion features for zero-shot text-driven motion transfer.

---

> > ### Comment · Reviewer_XgWS · 2025-08-05
> >
> > Thanks for the reply. The rebuttal has addressed most of my concerns. While I still have some concerns regarding the upper bound of zero-shot video editing in this era, the new perspective of unifying editing and sampling under continuous flow is valuable. I will increase my score accordingly.

---

### Official Review · Reviewer_77bu · 2025-07-02

**Clarity:** 2
**Significance:** 2
**Originality:** 3
**Rating:** 4
**Confidence:** 4

**Summary:**

The paper presents DFVEdit - a training-free video editing framework based on pre-trained Video DiTs.

Authors unify editing and sampling under the continuous flow perspective and propose to perform video editing by integrating proposed Conditional Delta Flow Vector (CDFV) as control term within the sampling process.

To further enhance the framework, the authors present Implicit Cross Attention (ICA), which functions as a CDFV masking mechanism. ICA facilitates DFVEdit and prevents local distributional drift in unedited regions.  Additionally, the authors propose Target Embedding Reinforcement, a technique that reweights text tokens in 3D Full-Attention. This approach amplifies the importance of text tokens most relevant to the editing task, improving the framework's ability to align with user-specified instructions.

DFVEdit is evaluated on real-world video datasets, including DAVIS2017 and Pexels, demonstrating superior performance compared to previous state-of-the-art methods.

**Questions:**

1. Please provide additional experiments that results for image editing, add results important baselines such as CogVideoX Video2Video pipeline and VidToMe, metrics for Wan model, and user study (Weakness 1).

2. Please provide method details (Weakness 2).

3.  In Algorithm 1, the initial latent sample is drawn only once. Could the quality of the edited video vary significantly with a different seed? What mechanism ensures stability and reproducibility across runs?

4. It seems that the approach appears to struggle with replacing or transforming objects of intricate shape (e.g., swapping a jeep for a tank or an elephant), suggesting limited capacity for large-scale geometric edits. Could you please elaborate on that?

**Ethical Concerns:**

["NO or VERY MINOR ethics concerns only"]

**Final Justification:**

The author addressed my major concerns, and I increase the rating to 4.

**Limitations:**

yes

**Quality:**

2

**Strengths And Weaknesses:**

## Strengths:

1. The method has a theoretical foundation

2. It is training-free and model-agnostic, making it highly versatile and widely applicable

3. The provided samples show good editing quality

## Weaknesses:

1. Insufficient experiments:

   (1) The authors did not provide user study results which is important for editing papers since current qualitative results only provide several selected generations which is not representative.

   (2) The proposed method is not specifically tied to video editing and can be tested on T2I models. I highly recommend authors to conduct image editing experiments since this field is more developed and has more developed and representative benchmarks (such as PIE-Bench [3]) as well as baseline methods (https://paperswithcode.com/sota/text-based-image-editing-on-pie-bench)

   (3) The paper misses important video editing baselines: CogVideoX Video2Video pipeline [1] and VidToMe [2].

   (4) Although DFVEdit achieves strong metrics, direct comparison with prior methods is complicated because they all rely on a U-Net backbone. Moreover, those baseline diffusion models produce relatively low-quality generations compared to modern DiT-based architectures (Wan2.1, CogVideoX). Therefore, a direct comparison is difficult to interpret.

   (5) The paper states that DEVEdit also works on Wan2.1-14B (line 218), however results are reported for CogVideoX only.


2. Lack of method details:
   (1) It is not clear which version of the DFVEdit procedure is actually used in the experiments - the simplified or the full variant. Do they yield different metric results?

   (2) The paper omits the analysis of the hyperparameter $\lambda$ (equations 6 and 31) and $\gamma$ (equation 16); how sensitive is the editing outcome to their choice?

   (3) The paper does not specify which attention blocks are used to extract the mask nor how the attention maps are binarized. This opacity suggests the masking may be suboptimal—background regions are reported to degrade during editing (as noted in “Limitations”).

[1] Yang, Z., Teng, J., Zheng, W., Ding, M., Huang, S., Xu, J., ... & Tang, J. (2024). Cogvideox: Text-to-video diffusion models with an expert transformer. arXiv preprint arXiv:2408.06072.

[2] Li, X., Ma, C., Yang, X., & Yang, M. H. (2024). Vidtome: Video token merging for zero-shot video editing. In Proceedings of the IEEE/CVF Conference on Computer Vision and Pattern Recognition (pp. 7486-7495).

[3] Ju, X., Zeng, A., Bian, Y., Liu, S., & Xu, Q. (2023). Direct inversion: Boosting diffusion-based editing with 3 lines of code. arXiv preprint arXiv:2310.01506.

---

> ### Author Rebuttal · Authors · 2025-07-29
>
> We sincerely thank you for your deep technical engagement with our work. Your meticulous analysis of experimental completeness and methodological details has significantly strengthened the paper's empirical foundations. Below are our responses to the concerns that have been raised.
>
> ### Weakness:
>
> **1. On Insufficient Experiments:**
>
> **(1) User Study:** We agree on the importance of user studies for evaluating subjective editing quality. We apologize for not highlighting the user study results before. As detailed in Table 1 (corresponding metric description in p. 9, L 252–255) and Appendix (p. 3, L 48–52), we conducted user studies with 20 participants assessing 80 video-prompt pairs across stylization, shape, and attribute editing (1,600 ratings in total). Results show DFVEdit outperforms baselines in video-prompt alignment ("Edit"), frame quality ("Quality"), and spatio-temporal consistency ("Consistency").
>
> **(2) Image Editing Experiments:** We appreciate the reviewer’s suggestion on evaluating image editing generalization. We conducted experiments on PIE-Bench [1] using Stable Diffusion 1.5, comparing DFVEdit with Instruct-pix2pix [2], PnP [3], and P2P [4] (PnP and P2P have adopted DirectInversion[1] techniques). Due to time constraints, we evaluated DFVEdit and T2I baselines on two representative subsets: *1_change_object_80* and *9_change_style_80*, using standard metrics—**Distance** (overall structure coherence), **PSNR/LPIPS/SSIM** (background preservation), and **CLIP-T** (alignment)—following DirectInversion [1]. Results in **Table R1** show the competitive performance and practical cross-modal generalization ability of DFVEdit. Our core contribution remains enabling efficient, zero-shot editing on modern **Video DiT models**, where DFVEdit overcomes the computational and architectural limitations of attention-based methods without requiring fine-tuning.
>
>
>
> *Table R1: Quantitative results on PIE-Bench [1] for image editing.*
>
> |        Method        | Structure | Background |        |        | Text Alignment |
> | :------------------: | :-------: | :--------: | :----: | :----: | :------------: |
> |                      | Distance↓ |   PSNR↑    | LPIPS↓ | SSIM↑  |    CLIP-T↑     |
> |       DFVEdit        |  0.0167   |   23.24    | 0.1225 | 0.6935 |     29.62      |
> | Instruct-pix2pix [2] |  0.0258   |   21.56    | 0.1308 | 0.6321 |     26.13      |
> |       P2P [4]        |  0.0186   |   22.38    | 0.1257 | 0.6416 |     29.07      |
> |       PnP [3]        |  0.0279   |   18.06    | 0.1353 | 0.6135 |     29.02      |
>
>
>
> **3) Additional Video Editing Baselines:**  We thank the reviewer for the suggestion. We have added comparisons with *CogVideoX-V2V* and *VidToMe* (with the same experiment settings in the comparison session). As shown in **Table R2**, DFVEdit outperforms both methods across all metrics. These results will be fully presented (including quantitative and qualitative) in the final version.
>
>
>
> *Table R2: Quantitative comparison with state-of-the-art video editing methods.*
>
> | Method                      | CLIP-F↑ | E_warp↓ | M.PSNR↑ | LPIPS↓ | CLIP-T↑ |
> | --------------------------- | ------- | ------- | ------- | ------ | ------- |
> | CogvideoxV2V [7]            | 0.9812  | 1.67    | 20.53   | 0.4092 | 27.46   |
> | VidToMe [8]                 | 0.9737  | 2.96    | 22.52   | 0.3062 | 27.32   |
> | **DFVEdit** (CogvideoX-5B)  | 0.9924  | 1.12    | 31.18   | 0.1886 | 30.84   |
> | **DFVEdit** (on Wan2.1-14B) | 0.9950  | 1.06    | 31.23   | 0.1568 | 31.34   |
>
>
>
> **(4) Comparison Fairness Across Architectures:**
>
> We thank the reviewer for the insightful comment. To ensure fair comparisons:
>
> - We evaluate SDEdit and CogVideoX-V2V on the **same Video DiT backbone** (CogVideoX-5B, Tables 1 and R2), where DFVEdit still outperforms.
> - We have conducted experiments on a **unified 2D U-Net** (Stable Diffusion 1.5) for image editing (Table R1), demonstrating the effectiveness of DFVEdit independent of Video-DiT-specific advantages.
>
> To the best of our knowledge, DFVEdit is the first method enabling efficient and effective zero-shot video editing on modern **Video DiTs**—including both score-matching (e.g., CogVideoX) and flow-matching (e.g., Wan2.1) models. Prior attention-based methods are often incompatible due to architectural and computational constraints. DFVEdit overcomes these via a lightweight, flow-field-based formulation.
>
> **(5) Results on Wan2.1-14B:** We have included *qualitative results* using Wan2.1-14B in Figure 4 (p. 8) and Figure F5 (Appendix, p. 10). For completeness, we now provide *quantitative results* in **Table R2**, showing that DFVEdit achieves better performance when built upon this stronger Video DiT backbone.
>
> Thank you for all your suggestions on experiments. Visualization results, however, can not be shown in this rebuttal phase due to rebuttal policy constraints. **We will update all qualitative visualization results in the final version.**
>
>
>
> **2. On Lack of Method Details**
>
> **(1) DFVEdit Version:** We used the full version of DFVEdit, including both the ICA and ER components. The effectiveness of these components is validated in the ablation studies (main text, §4.3), and further implementation details are provided in Appendix B.3, including Figure F1 and Figure F2, which illustrate ER ablation on shape editing and ICA extraction and visualization.
>
> **(2) Hyperparameter Selection:** For Eq. 16, we conducted ablation studies (Appendix B.3, Figure 5a and Figure F1) to determine optimal values. These hyperparameters are stable and easy to tune, typically requiring only a single sweep across a small range. The parameters in Eq. 6 and Eq. 31 relate to the theoretical unification of editing and sampling dynamics. In practice, DFVEdit introduces only a few key hyperparameters: the fusion timesteps for ICA and the number of sampling iterations (T).
>
> **(3) Mask Usage:** The mask is applied in Layer 16's cross-attention and is timestep-aware, following FreeMask's design, evolving from coarse to fine during sampling. This allows objects greater flexibility in shape deformation, leading to more natural editing boundaries compared to rigid masks (e.g., in VideoGrain or VideoDirector). While this may cause minor leakage into background regions, resulting in subtle background changes, our CDFV mechanism and ICA module effectively constrain such drift. In most cases, background preservation remains strong.
>
>
>
>
>
> ### Questions
>
> **Q1&Q2 Additional Experiments and Method Details**: Please refer to the weaknesses.
>
> **Q3. Seed Sensitivity and Reproducibility:**  The latent initialization stage is noise-free, meaning we transform a clean source video latent into a clean target latent. When computing the Conditional Delta Flow Vector (CDFV), the *same* Wiener process $dW$ ( the same random seeds) is applied to both $\hat{Z}_t$ (target latent at t) and $Z_0$ (source latent) In the forward flow (Eq. 1), which leads to the cancellation of stochastic noise in the output difference (Eq. 12). Additionally, each forward pass uses a new random seed, preventing bias accumulation from fixed noise patterns.  We tested DFVEdit (on CogVideoX-5B) on 20 video-prompt pairs with five different repetitive experiments to evaluate stability across random seeds. Table R3 reports the mean and standard deviation of key metrics, indicating low variance and consistent performance. The stability arises from our CDFV design—only the *deterministic* score difference drives editing, and ICA further enhances consistency by anchoring background features.
>
>
>
> *Table R3: Performance stability across 5 repetitive experiments (mean ± std).*
>
> | Method  | CLIP-F↑      | E_warp↓   | M.PSNR↑  | LPIPS↓      | CLIP-T↑  |
> | ------- | ------------ | --------- | -------- | ----------- | -------- |
> | DFVEdit | 0.9924±0.003 | 1.11±0.12 | 31.2±1.3 | 0.189±0.012 | 30.8±0.9 |
>
>
>
> **Q4. Limitations in Large-Scale Geometric Editing:** We agree with the reviewer that DFVEdit, like most text-driven editing methods, struggles with large-scale geometric transformations (e.g., replacing a jeep with a tank or an elephant). Our method operates via a delta flow vector that models smooth transformations between source and target videos. It excels at stylization, attribute editing (color, texture), local shape changes, and substitutions with similar structure (e.g., car → truck), but is less suited to radical topological changes. This limitation stems from the flow-based formulation and the underlying DiT's reliance on source structure. Generating entirely new, complex geometries while preserving background and motion remains a fundamental challenge in video editing. We focus on high-fidelity, temporally consistent semantic editing with minimal user input. Future work may incorporate explicit structural guidance (e.g., depth, optical flow, layout maps) or layered compositing strategies for extreme shape changes.
>
>
>
> [1] Ju, X., Zeng, A., Bian, Y., Liu, S., & Xu, Q. (2023). Direct inversion: Boosting diffusion-based editing with three lines of code. arXiv preprint arXiv:2310.01506.
>
> [2] Brooks, T., Holynski, A., & Efros, A. A. (2023). Instructpix2pix: Learning to follow image editing instructions.
>
> [3] Tumanyan, N., Geyer, M., Bagon, S., & Dekel, T. (2023). Plug-and-play diffusion features for text-driven image-to-image translation.
>
> [4] Hertz, A., Mokady, R., Tenenbaum, J., Aberman, K., Pritch, Y., & Cohen-Or, D. (2022). Prompt-to-prompt image editing with cross attention control.

---

> > ### Comment · Reviewer_77bu · 2025-08-07
> >
> > Dear Authors,
> >
> > Thank you for your clarifications and additional experiments. My major concerns are addressed and I increase my rating to 4. I recommend to include information from your rebuttal to camera-ready version of the paper as it will enhance the paper's clarity and overall quality.
> >
> > Regarding minor comments, there is some inconsistency between image editing metrics reported in Table R1 and those presented in the PnP-Inversion paper. It is important to resolve this discrepancy to ensure the accuracy and reliability of your results. In addition, I highly recommend recent image editing methods that utilize Flux and SD3 models.

---

### Comment · Area_Chair_Sa8G · 2025-08-04
**Request for your feedback in light of authors' feedback**

Thank you for your valuable insights and expertise which have contributed significantly to the review process.

Following the initial review, the authors have provided a detailed rebuttal addressing the feedback and comments provided by our esteemed reviewers, including yourself. I kindly request that you take the time to carefully review the authors' rebuttal and assess its impact on your initial evaluation.

Please share your thoughts and any additional points you may have after reading the authors' rebuttal. Thank you very much!

---

### Note · Authors · 2025-08-14

**Dear Area Chairs and Reviewers,**

We sincerely thank all reviewers for their thoughtful feedback and the Area Chairs for facilitating this discussion. We are grateful that Reviewers 77bu and XgWS have revised their scores upward following our rebuttal. Below, we summarize key revisions and core contributions.

**Response to Reviewer 77bu:** We highlighted user study details, extended image editing experiments on SDv1.5 (PIE-Bench), added comparisons with VidToMe/CogVideoX V2V, and included Wan2.1 results. *Additionally, our work has now been extended to image editing on FLUX and SD3 (will be presented in the final version). Most Table R4 metrics align with Direct Inversion's Table 1, except that 'CLIP-T' is renamed to 'CLIP Similarity' (Whole) for consistency.*

**Response to Reviewer XgWS:** We clarified DFVEdit's zero-shot motivation beyond efficiency. We demonstrated diverse editing tasks and experimental scope and ensured fair baseline comparisons.

**Response to Reviewer Q6ss:** We clarified DFVEdit's non-attention-based mechanism and enhanced experimental rigor via motion complexity and ablation results analysis. Figure 2 annotations were improved, token identification and aspect ratio flexibility were clarified, and the VAE bottleneck was discussed in the limitations.

**Response to Reviewer EYj3:** We conducted diffusion step ablation, highlighted DFVEdit's dynamic consistency superiority over VideoGrain, and distinguished ICA from attention engineering. Relevant citations expanded on limitations on non-rigid editing and drag-based interaction.

**Core Contributions of DFVEdit:**

1. **CDFV Framework**: Unifies sampling and editing via continuous flow transformations, revealing editing as a conditional latent flow field shift.
2. **SOTA Performance**: Achieves state-of-the-art structure fidelity, temporal consistency, and editing quality across CogVideoX, Wan2.1.
3. **Model-Agnostic Operation**: Works on 2D U-Net and Video DiTs architectures without fine-tuning or architecture-specific modifications.
4. **Efficiency & Scalability**: 20× faster inference, 85% memory reduction vs. attention-engineering-based methods; supports long sequences (>20 frames) and non-1:1 aspect ratios.

All revisions will be integrated into the camera-ready version. The manuscript addresses all concerns and presents a robust, theoretically grounded framework for zero-shot video editing. Thank you for your time and consideration.

**Sincerely,**

**The Authors**

---

### Decision · Program_Chairs · 2025-09-17

**Decision:**

Reject

**Comment:**

This paper presents DFVEdit, a training-free framework for video editing with Video DiTs. The method introduces Conditional Delta Flow Vector (CDFV) together with Implicit Cross Attention and Embedding Reinforcement to unify editing and sampling under the flow perspective. The approach is versatile, model-agnostic, and shows efficiency gains with encouraging editing quality.

The main weakness lies in clarity, which prevented reviewers from giving a firm assessment. All four reviewers rated clarity as 1 or 2, and I stand by their detailed questions and suggestions. Another concern is evaluation: several results focus on simple style edits rather than challenging cases, important baselines are missing, and the framework appears complex without clear evidence of benefit. The limited upper bound of training-free methods also remains a concern.

Overall, while the idea is interesting and technically grounded, reviewers were not convinced due to unclear presentation and insufficient evaluation. Since no reviewer strongly championed the paper, I recommend rejection.